# Solution and crystal structures of a C-terminal fragment of the neuronal isoform of the polypyrimidine tract binding protein (nPTB)

Amar Joshi[1,3], Vicent Esteve[2,3], Adrian N. Buckroyd[1], Markus Blatter[2], Frédéric H.-T. Allain[2] and Stephen Curry[1]

[1] Department of Life Sciences, Imperial College, London, United Kingdom
[2] Institute of Molecular Biology and Biophysics, ETH Zürich, Switzerland
[3] These authors contributed equally to this work.

## ABSTRACT

The eukaryotic polypyrimidine tract binding protein (PTB) serves primarily as a regulator of alternative splicing of messenger RNA, but is also co-opted to other roles such as RNA localisation and translation initiation from internal ribosome entry sites. The neuronal paralogue of PTB (nPTB) is 75% identical in amino acid sequence with PTB. Although the two proteins have broadly similar RNA binding specificities and effects on RNA splicing, differential expression of PTB and nPTB can lead to the generation of alternatively spliced mRNAs. RNA binding by PTB and nPTB is mediated by four RNA recognition motifs (RRMs). We present here the crystal and solution structures of the C-terminal domain of nPTB (nPTB34) which contains RRMs 3 and 4. As expected the structures are similar to each other and to the solution structure of the equivalent fragment from PTB (PTB34). The result confirms that, as found for PTB, RRMs 3 and 4 of nPTB interact with one another to form a stable unit that presents the RNA-binding surfaces of the component RRMs on opposite sides that face away from each other. The major differences between PTB34 and nPTB34 arise from amino acid side chain substitutions on the exposed $\beta$-sheet surfaces and adjoining loops of each RRM, which are likely to modulate interactions with RNA.

Corresponding authors
Frédéric H.-T. Allain,
allain@mol.biol.ethz.ch
Stephen Curry,
s.curry@imperial.ac.uk

## INTRODUCTION

Alternative splicing is the rule rather than the exception in the human genome. Over 95% of multi-exon genes produce alternatively spliced mRNAs (*Nilsen & Graveley, 2010*), yielding many more protein variants or isoforms than specified by the number of genes. The process of alternative splicing is controlled by regulatory sequences within pre-mRNA transcripts that recruit suites of proteins to determine whether selected exons are included or excluded as mRNA is brought to maturity.

The polypyrimidine tract binding protein (PTB) is a well-characterized regulator of alternative splicing (*García-Blanco, Jamison & Sharp, 1989*; *Gil et al., 1991*) although it also

has significant roles that affect the processing, localization and use of a variety of mRNAs (reviewed by *Sawicka et al., 2008*). PTB does not recognize a unique RNA sequence; rather it is specific for short motifs (e.g., UCUCU, UCUU) within pyrimidine rich regions of pre-mRNA (*Singh, Valcárcel & Green, 1995*; *Pérez et al., 1997*). In alternative splicing, PTB binding is most commonly associated with exclusion or skipping of the regulated exon, although more recent genome-wide splicing studies have indicated that PTB binding can also promote exon *inclusion* in a minority of cases (*Xue et al., 2009*; *Llorian et al., 2010*; *Witten & Ule, 2011*).

Three tissue-specific homologues of PTB have been identified—smPTB, ROD1 and nPTB—all of which have high levels of amino acid sequence identity with the prototypical protein (69–74%) and are therefore likely to have very similar structures. smPTB (smooth muscle tissue) has so far only been found in rodents (*Gooding, Kemp & Smith, 2003*), whereas the homologue ROD1 is expressed primarily in hematopoietic tissues (*Yamamoto et al., 1999*). nPTB is found predominantly in brain and testis but also, at low levels, in muscle cells (*Markovtsov et al., 2000*; *Polydorides et al., 2000*).

PTB and nPTB bind to the same RNA sequences and have similar effects on alternative splicing events for a number of transcripts (*Markovtsov et al., 2000*; *Pilipenko et al., 2001*; *Spellman, Llorian & Smith, 2007*). However, for some neuronal-specific alternative splicing events the expression of nPTB leads to distinct outcomes. For example, the tyrosine kinase *c-src* has an *n-src* isoform found only in brain tissue that requires nPTB for the selective inclusion of a neuronal-specific exon (*Markovtsov et al., 2000*; *Sharma, Falick & Black, 2005*; *Boutz et al., 2007*).

PTB (and its homologues) contain 4 RNA recognition motif domains (RRMs) separated by long flexible linkers and arrayed, when not bound to RNA, in a relatively extended, linear conformation (*Petoukhov et al., 2006*). The solution structures of each RRM have been solved in the absence (*Conte et al., 2000*; *Simpson et al., 2004*; *Vitali et al., 2006*) and presence (*Oberstrass et al., 2005*) of RNA. While RRMs 1 and 4 adopt a canonical $\beta\alpha\beta\beta\alpha\beta$ topology with two helices packed against a four-strand $\beta$-sheet that forms the primary RNA binding surface, RRMs 2 and 3 have a modified architecture. In these domains a C-terminal extension beyond $\beta4$ loops across the upper edge of the $\beta$-sheet and adds a fifth strand on the far side of the sheet; this expands the size of the RNA binding platform, while the $\beta4$-$\beta5$ loop provides additional points of interaction with RNA (*Conte et al., 2000*; *Simpson et al., 2004*; *Oberstrass et al., 2005*).

The $\alpha$-helical regions of RRMs 3 and 4 pack together to form a stable unit in which the RNA-binding $\beta$-sheets are exposed on opposite sides; it has been proposed that this configuration enforces looping of RNA that plays an important role in defining exon structures to be excised from pre-mRNA (*Oberstrass et al., 2005*; *Vitali et al., 2006*; *Lamichhane et al., 2010*). Although early studies had suggested that RRM3 and RRM4 might not interact at physiological salt concentrations (*Conte et al., 2000*; *Clerte & Hall, 2006*), it is now generally accepted that this pair of RRMs have fixed relative orientations (*Lamichhane et al., 2010*; *Maynard & Hall, 2010*). Evidence suggests that RRM1 and RRM2 bind RNA as individual domains although RRM2 is also involved in protein–protein
interactions with the splicing co-regulator Raver1 (*Rideau et al., 2006*). Crystallographic analysis has revealed how the helical face of the PTB RRM2 provides a binding surface for specific recognition of specific peptide motifs in Raver1 (*Joshi et al., 2011*).

Although PTB and nPTB are likely to have similar structures because of their high sequence identity (75%) (Fig. 1A), to begin to tease out the structural basis for observed differences in splicing regulation we have determined the structure of a C-terminal fragment of nPTB that contains RRMs 3 and 4 (nPTB34). We have solved the structures both crystallographically and by NMR. Here we compare those structures with each other and with the solution structure of the equivalent fragment from PTB isoform1 (called here PTB34).

## RESULTS AND DISCUSSION

### Crystal structure of nPTB34

nPTB34, a C-terminal nPTB fragment that contains RRMs 3 and 4 (residues 336-531) was over-expressed in *E. coli*, purified and crystallized by sitting drop vapor diffusion in 20% PEG 6000, 0.1 Tris pH 8.0 with 2 mM $ZnCl_2$ as an additive (see Materials and Methods). The crystals diffracted X-rays to 1.7 Å and were found to belong to space-group $P1$. Attempts to phase the diffraction data by molecular replacement using the solution structure of PTB34 (*Vitali et al., 2006*) were unsuccessful. Instead, phases were obtained by multi-wavelength anomalous dispersion (MAD) using data collected from a crystal of Se-Met-labeled nPTB34. This produced a high-quality electron density map that revealed a total of eight nPTB34 molecules in the asymmetric unit (Fig. 1B). The eight polypeptide chains were built almost in their entirety; due to poor electron density residue 336 at the start of nPTB34 was omitted, as were residues 420-423 (and one or two flanking residues in some chains) at the C-terminal end of the $\beta4$-$\beta5$ loop of RRM3. The final model, which incorporates 788 water molecules, was refined to an $R_{free}$ of 20.8% with good stereochemistry (Fig. 1C). Full data collection and refinement statistics are given in Table 1.

The structures of the eight copies of nPTB34 within the asymmetric unit of our crystals are very similar to one another (Fig. 1D). Pairwise superpositions of the eight chains give an average root-mean-square deviation (RMSD) for $C_\alpha$ atoms of only 0.17 Å. This is not surprising since the unit cell parameters deviate only marginally from a $P2_12_12$ space group (see Materials and Methods); what this means is that there are only two positions in the asymmetric unit, exemplified by chains A, B, E and F on the one hand and C, D, G and H on the other, that have significantly different packing environments.

There are a total of 12 $Zn^{2+}$ ions associated with each asymmetric unit which mediate crystal contacts between nPTB34 molecules involving variously His 412, His 491 and His 520. In each case the required tetrahedral coordination appears to be completed by $Cl^-$ ions and/or water molecules. These zinc-mediated interactions explain why the cation is required to obtain crystals (see Materials and Methods) but, since only two of the four ligands are provided by amino acids from nPTB34, they are not considered to be physiologically significant.

a

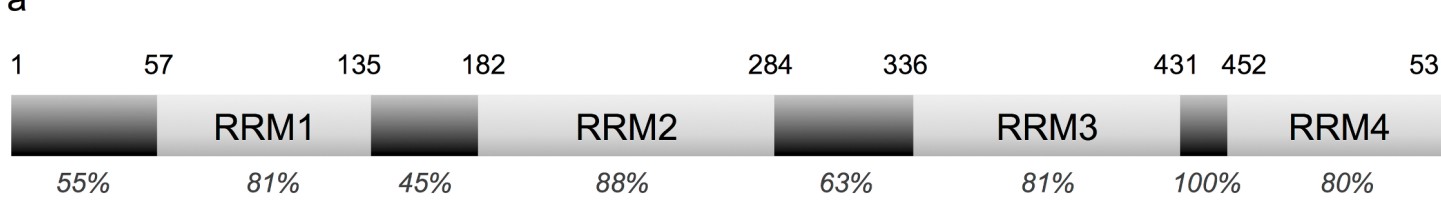

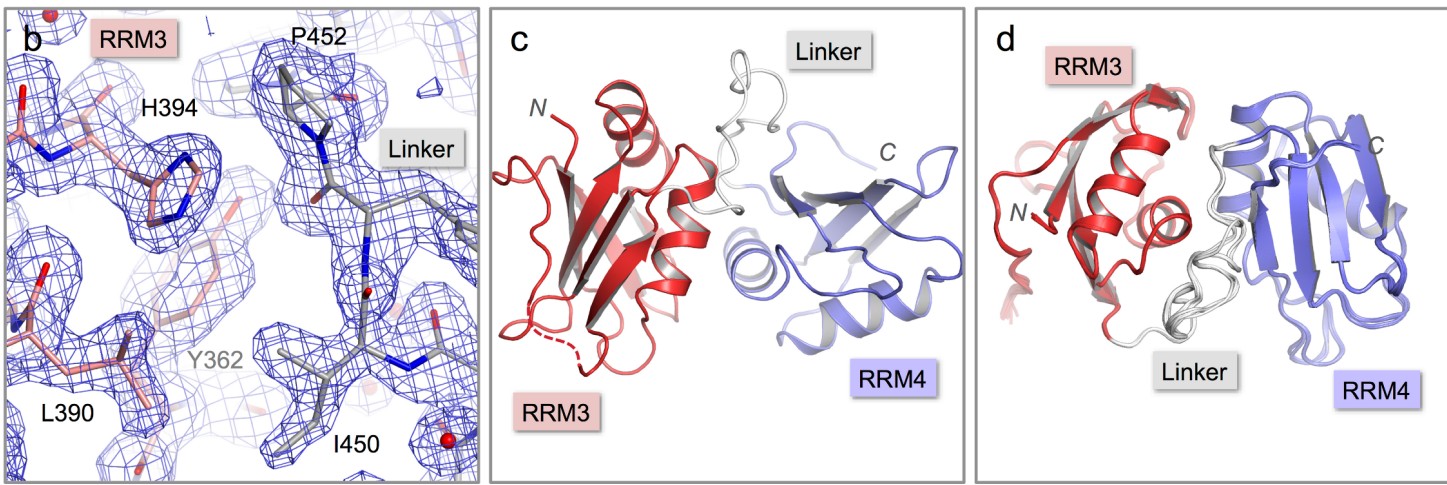

**Figure 1 Crystal structure of nPTB34.** (A) Domain structure of nPTB. Residue numbers for domain boundaries are given along the top; percentages below indicate the sequence identity with PTB within each defined domain or linker region. (B) $3F_o-2F_c$ electron density map contoured at $1\sigma$ for the refined crystal structure of nPTB34 (shown in stick representation). Carbon atoms in RRM3, RRM4 and the inter-domain linker are coloured pink, light-blue and grey respectively, a colour-scheme that is maintained throughout for nPTB34; nitrogen and oxygen atoms are coloured blue and red respectively. (C) The overall fold of the crystal structure of nPTB34 showing secondary structure features. (D) Alignment of all eight chains within the crystal asymmetric unit. View rotated with respect to panel C showing good overall alignment of secondary structure elements but differences in the linker.

## NMR structure of nPTB34

The solution NMR structure of nPTB34 was obtained from a construct containing residues 325–531 and is very similar to the one used for crystallography (see Materials and Methods). Using a total of 3622 nOe-derived distance restraints and 55 hydrogen bond restraints identified from slow exchanging amides we derived a fairly good ensemble of nPTB34 conformations in solution with a heavy atom RMSD of 0.71 Å of the structured region (Fig. 2A; Table 2). None of the restraints were violated by more than 0.3 Å and 67% of backbone torsions of the structured residues lay in the most favored region defined by the Ramachandran plot. Due to signal overlaps and broad linewidths the $\beta$2-$\beta$3-loops of both RRMs, the $\beta$4-$\beta$5 loop RRM3 and part of the second beta strand in RRM3 could not be completely assigned. Missing assignments in $\beta$2 of RRM3 appear to be the reason for the slightly different packing of the first alpha helix ($\alpha$1) in RRM3 against the $\beta$-sheet compared to the crystal structure. Since $\alpha$1 is part of the domain interface this different packing results also in a slight difference in the domain interface

**Table 1 Data collection and refinement statistics for the crystal structure of nPTB34.**

| Space-group | $P1$ | | $P2_1$ | |
|---|---|---|---|---|
| **Data collection** | | | | |
| $a, b, c$ (Å) | 61.00, 65.81, 99.58 | | 65.81, 60.95, 99.55 | |
| $\alpha, \beta, \gamma$ (°) | 89.99, 90.00, 89.99 | | 90.00, 90.01, 90.00 | |
| | | **Peak** | **Inflection** | **Remote** |
| Wavelength (Å) | 0.9795 | 0.9808 | 0.9813 | 0.9790 |
| Resolution range (Å) | 44.74–1.69 | 55.10–2.30 | 55.14–2.30 | 55.27–2.50 |
| | (1.72–1.69) | (2.42–2.30) | (2.42–2.30) | (2.64–2.50) |
| No. of independent reflections | 153 732 (7340) | - | - | - |
| Multiplicity[a] | 1.9 (1.9) | 1.7 (1.6) | 1.7 (1.7) | 1.7 (1.7) |
| Completeness (%) | 89.0 (85.6) | 89.9 (84.8) | 95.9 (96.1) | 95.7 (93.5) |
| $I/\sigma_I$ | 5.0 (1.1) | 10.0 (4.3) | 8.7 (3.0) | 8.1 (2.4) |
| $R_{merge}$ (%)[b] | 9.6 (70.7) | 6.7 (21.2) | 8.6 (37.9) | 9.4 (47.8) |
| **Model refinement** | 40.81–1.69 Å | | | |
| No. of non-hydrogen atoms/waters | 12 397 | | | |
| $R_{work}$ (%)[c] | 17.00 | | | |
| $R_{free}$ (%)[d] | 20.76 | | | |
| RMS bonds (Å)[e] | 0.006 | | | |
| RMS bond angles (°) | 0.937 | | | |
| Ramachandran plot (% favoured/allowed) | 97.4/1.9 | | | |
| PDB Identifier | 4cq1 | | | |

**Notes.**

[a] Values for highest resolution shell given in parentheses.

[b] $R_{merge} = 100 \times \Sigma_{hkl}|I_j(hkl) - \langle I_j(hkl)\rangle|/\Sigma_{hkl}\Sigma_j I(hkl)$, where $I_j(hkl)$ and $\langle I_j(hkl)\rangle$ are the intensity of measurement $j$ and the mean intensity for the reflection with indices $hkl$, respectively.

[c] $R_{work} = 100 \times \Sigma_{hkl}||F_{obs}| - |F_{calc}||/\Sigma_{hkl}|F_{obs}|$.

[d] $R_{free}$ is the $R_{work}$ calculated using a randomly selected 5% sample of reflection data that were omitted from the refinement.

[e] RMSD, root-mean-squared deviations.

compared to the nPTB34 crystal structure and RNA bound and free PTB34 solution structures.

## Comparison of crystal and NMR structures of nPTB34

Overall there is very good correspondence between the structures of nPTB34 determined crystallographically and by NMR. Superposition of the two models using just $C_\alpha$ atoms gives an RMSD of 3.0 Å, and RMSDs of 1.8 and 2.6 Å for RRMs 3 and 4 respectively (Fig. 2B). These values are slightly higher than for the superposition of the crystal structure of nPTB34 onto the solution structures of PTB34 in the absence or presence of RNA, which both give overall RMSDs of 1.3 Å (Fig. 2C). This largely reflects the fact that the solution structure of nPTB34 (unlike PTB34 (*Vitali et al., 2006*)) was determined without the aid of segmental labeling and thus chemical shift completeness for protons of only 77% could be achieved. The relative orientation of RRMs 3 and 4 differs slightly between

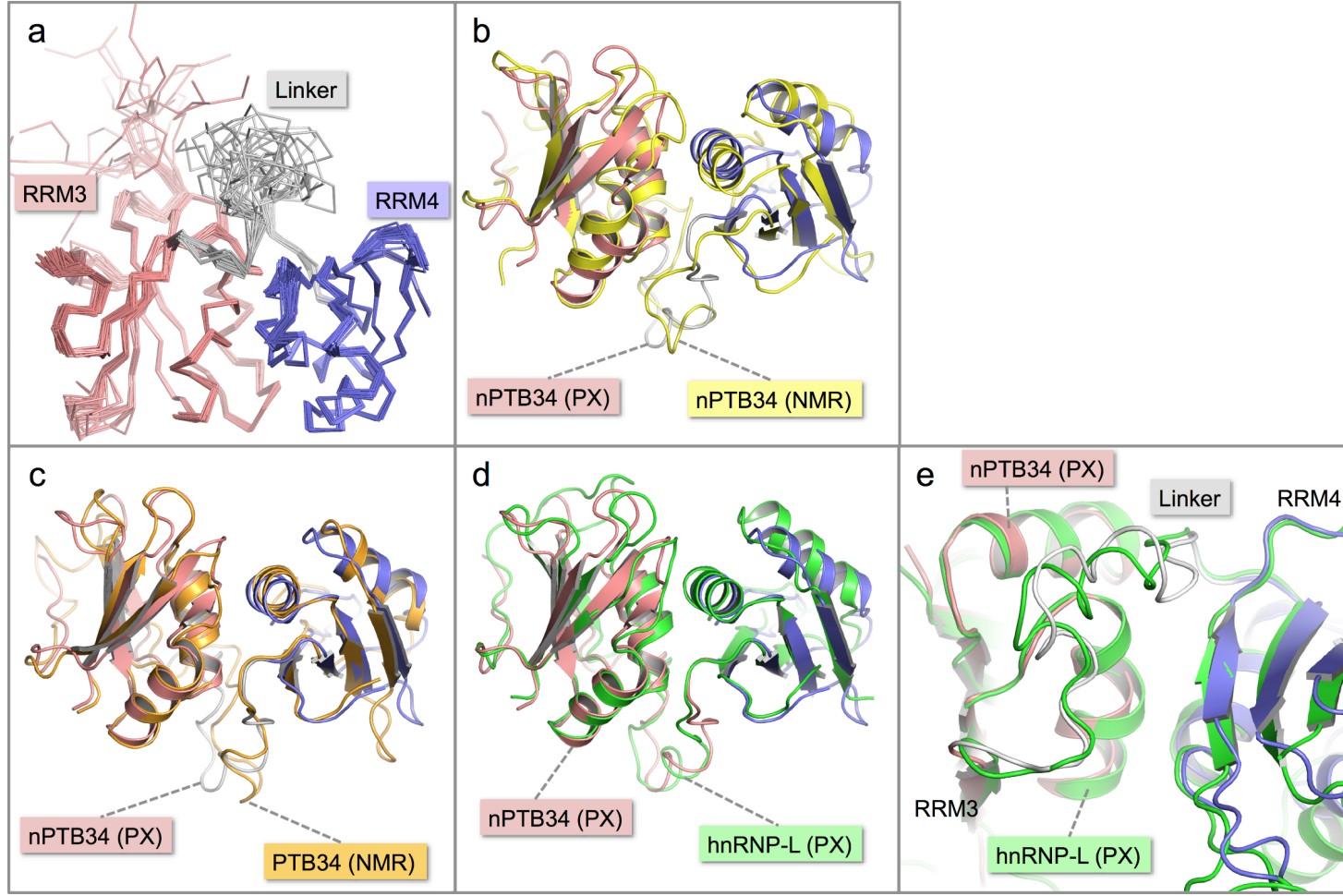

**Figure 2 Solution structure of nPTB34 and comparison with related structures.** (A) Ensemble of the 20 lowest energy solution structures of nPTB34 shown as $C_\alpha$ traces coloured by domain. (B) Ribbon representation of the superposition of the crystal structure of nPTB34 (coloured by domain) on the nPTB34 solution structure (yellow) denoted by 'PX' and 'NMR' labels respectively. (C) Superposition of the crystal structure of nPTB34 (coloured by domain) on the solution structure of PTB34 (PDB ID 2evz; orange) (*Vitali et al., 2006*). (D) Superposition of the crystal structure of nPTB34 (coloured by domain) on the crystal structure of the equivalent domain from hnRNP-L (PDB ID 3 to 8; green) (*Zhang et al., 2013*). (E) Close up of the superposition shown in panel D to illustrate the similarities and differences in the linker regions between RRMs 3 and 4 in nPTB34 and hnRNP-L.

the crystal and solution structures of nPTB34 but this is unlikely to be significant since there is no significant difference in the domain orientations between the crystal structure of nPTB34 and the solution structure of PTB34 (see below); moreover superposition of the crystal structures of the PTB34 fragments from nPTB and hnRNP-L (*Zhang et al., 2013*) (RMSD = 1.3 Å) shows that the relative disposition of RRM3 and RRM4 is highly conserved between these two proteins (Fig. 2D).

Within each RRM of nPTB34 there is good correspondence of secondary structure features between the crystal and solution structures. Those loops in the crystal structure with the highest temperature factors (RRM3: $\beta2$-$\beta3$ and the C-terminal end of $\beta4$-$\beta5$; RRM4: $\beta1$-$\alpha2$, $\beta2$-$\beta3$, $\alpha2$-$\beta4$; and the inter-domain linker) are those exhibiting greatest mobility in the solution structure (Figs. 2A and 2B). Notably, the $\beta4$-$\beta5$ loop is not

| Table 2 Statistics for the solution structure of nPTB34. | |
| --- | --- |
| Number of distance restraints | 3677 |
| Intraresidual | 657 |
| Sequential ($|i-j| = 1$) | 941 |
| Medium range ($1 < |i-j| < 5$) | 684 |
| Long range ($|i-j| \geq 5$) | 134 |
| Hydrogen bonds[a] | 55 |
| Energy statistics[b] | |
| Average distance constraint violations (Å) | |
| 0.1–0.2 Å | $52.4 \pm 4.2$ |
| 0.2–0.3 Å | $0.4 \pm 0.6$ |
| >0.3 Å | $0.0 \pm 0.0$ |
| Maximal (Å) | $0.20 \pm 0.03$ |
| Mean AMBER distance violation energy (kcal mol$^{-1}$) | $100.9 \pm 2.5$ |
| Mean AMBER energy (kcal mol$^{-1}$) | $-6170.0 \pm 15.5$ |
| Mean deviation from ideal covalent geometry | |
| Bond length (Å) | $0.0042 \pm 0.0000$ |
| Bond angle (degrees) | $1.796 \pm 0.007$ |
| Ramachandran plot statistics[b,c,d] | |
| Most favoured regions (%) | $67.1 \pm 2.0$ |
| Additionally allowed regions (%) | $29.2 \pm 1.9$ |
| Generously allowed regions (%) | $3.5 \pm 0.8$ |
| Disallowed regions (%) | $0.1 \pm 0.2$ |
| RMS deviations from mean structure Statistics[b,c] | |
| Backbone atoms (Å) | $0.37 \pm 0.05$ |
| Heavy atoms (Å) | $0.71 \pm 0.05$ |
| PDB identifier | 2mju |

**Notes.**
[a] Hydrogen bond constraints were identified from slow exchanging amide protons in $D_2O$.
[b] Statistics computed for the deposited bundle of 20 violation energy best structures selected out of 30 energy best structures refined in Amber (*Pearlman et al., 1995*).
[c] Based on structured residue range as defined by cyana command overlay: 9–47, 52–86, 91–113, 126–163, 168–207.
[d] Ramachandran plot, as defined by the Procheck (*Laskowski et al., 1996*).

disordered in the crystal structure of RRM2 but this may simply be because it is involved in more crystal contacts (*Joshi et al., 2011*).

Within the inter-domain linker (residues 433–453), the central portion (residues 439–448) exhibits considerable conformational variation in the solution structure of nPTB34 (Fig. 2A). A similar but slightly smaller region (residues 443–449), which protrudes away from the body of the domain, is variable between the eight chains present in the asymmetric unit of the nPTB34 crystals (Fig. 1D). Again, this likely reflects packing variations since there are effectively only two distinct conformers evident for residues 443–449, which partition into the two distinct packing environments of the pseudo $P2_12_12$ space group (see above). The N- and C-terminal portions of the linker are involved in a number of specific polar and nonpolar interactions that help to stabilize the packing of RRM3 and RRM4 and are described in more detail below.

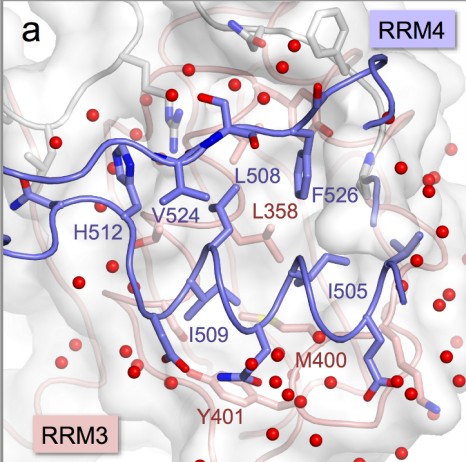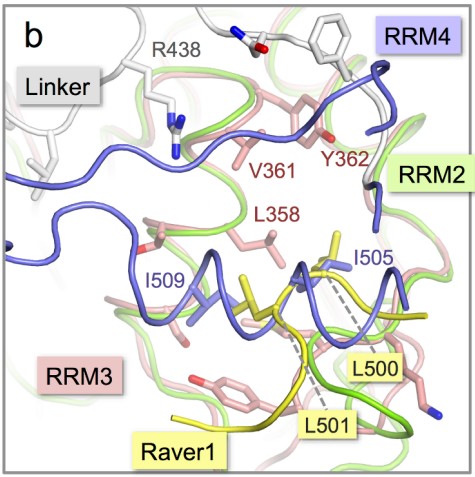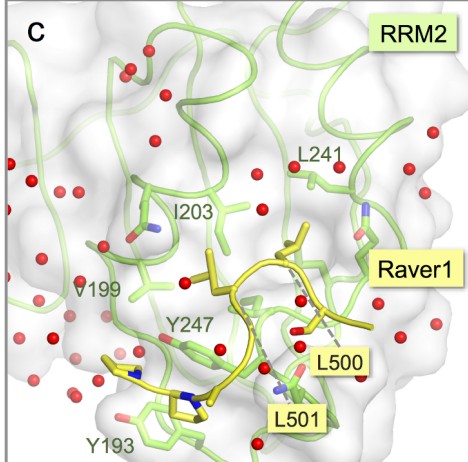

**Figure 3 Comparison of the inter-domain interface within nPTB34 with the PTB RRM2-Raver1 interaction.** (A) The inter-domain interface in the crystal structure of nPTB34. RRM3 is shown with pink backbone and side-chain carbon atoms along with a semi-transparent rendering of its molecular surface (light-grey). The portion of RRM4 that contacts RRM3 within nPTB34 is shown without its molecular surface. The oxygen atoms of bound water molecules located in the crystal structure are indicated by red spheres. (B) Superposition of the nPTB34 crystal structure (coloured as in panel A) with the structure of the PTB RRM2-Raver1 complex (PDB ID 3zzy; RRM2—green; Raver1—yellow) (*Joshi et al., 2011*). Selected side chains are indicated. (C) The interaction between PTB RRM2 (green) and Raver1 (yellow). As for RRM3 in panel A, the molecular surface of RRM2 is indicated by a semi-transparent rendering.

## Comparison of nPTB34 with PTB34

PTB and nPTB have 75% sequence identity overall but this is focused in the RRMs where the sequence identity between equivalent domains ranges from 80% (RRMs 3 and 4) to 88% (RRM2) (Fig. 1A). As expected, the three-dimensional structures of nPTB34 and PTB34 are very similar; differences are largely confined to the local effects of amino acid substitutions.

The crystal structure of nPTB34 reveals that the interaction between RRM3 and RRM4 is stabilized by the exclusion of water molecules, which allows direct contact between a conserved set of hydrophobic residues—Ile 505, Leu 508, Ile 509 and Phe 526 from RRM4 and the apolar portions of Ser 354, Thr 357, Leu 358, Val 361 and Met 400 from RRM3 (Fig. 3A). Ringing this hydrophobic core are direct and water-mediated hydrogen bonds between the RRMs and from the RRMs to the linker polypeptide. Within the RRM3-RRM4 interface there are only five amino acid changes between PTB34 and nPTB34 and these are generally of a conservative character; the apolar residues Ile 356, Met 400 and Val 505 in PTB34 are replaced by Thr 357, Leu 399 and Ile 505 respectively in nPTB34, while His 397 and His 400 in PTB34 are altered to Gln 398 and Tyr 401 (Table 3). The water-mediated hydrogen bond interaction from Tyr 401 of RRM3 to the side chain of Asp 510 in RRM4 of nPTB34 is therefore lost in PTB34 but neither this nor the other amino acid changes appear sufficient to significantly alter the interaction between the two domains. A similar degree of conservation of the apolar residues at the hydrophobic interface between RRM3 and RRM4 is also seen in hnRNP-L (*Zhang et al., 2013*) (Table 3). The relative orientation of these domains is therefore a conserved feature of PTB, nPTB and hnRNP-L. The incorporation of a pair of His residues within the inter-domain interface of PTB34 (H397, H400) may

**Table 3 Conservation of amino acids within the interface between RRM3-RRM4 in PTB, nPTB and hnRNP-L.**

|      | PTB  | nPTB | hnRNP-L |
|------|------|------|---------|
| **RRM3** | S353 | S354 | R398 |
|      | I356 | T357 | N401 |
|      | L357 | L358 | V402 |
|      | V360 | V361 | L405 |
|      | Y361 | Y362 | Y406 |
|      | H397 | Q398 | N442 |
|      | L399 | M400 | M444 |
|      | H400 | Y401 | F445 |
| **RRM4** | V501 | V501 | K552 |
|      | V505 | I505 | L556 |
|      | L508 | L508 | L559 |
|      | I509 | I509 | G560 |
|      | F526 | F526 | F582 |

affect the stability of the module at low pH (*Nordlund et al., 2003*), but we are not aware of any circumstances where pH regulation of the conformation would be applied.

Although the interaction of RRM4 with the helical face of RRM3 within PTB34 and nPTB34 is much more extensive than the interaction of PTB-binding peptides in Raver1 with the equivalent face of RRM2 (*Joshi et al., 2011*), there is one interesting parallel. Superposition of the RRM2-raver1 co-crystal structure onto RRM3 of nPTB34 reveals that the Raver1 peptide occupies essentially the same position as helix $\alpha$2 of RRM4 (Figs. 3B and 3C). Although the Raver1 peptide is non-helical—in fact it has a pinched but largely extended conformation—it inserts a pair of Leu residues (Leu 500 and Leu 501 in PDB entry 3zzy) into a hydrophobic pocket formed between the helices $\alpha$1 and $\alpha$2 of RRM2 that effectively mimics the positioning of Ile 505 and Ile 509 from RRM4 into the equivalent pocket on RRM3 within nPTB34.

Although the RRMs are the most highly conserved features between PTB and nPTB, there are some notable differences on their RNA binding surfaces. These differences appear to be more concentrated on RRM3. While there are a total of five amino acid substitutions in nPTB RRM3 of residues identified to contact RNA in the solution structure of the complex of PTB34 with a CUCUCU oligonucleotide (*Oberstrass et al., 2005*), there are only two such amino acid changes in RRM4 (Fig. 4).

The replacement of Phe 371 in the $\beta$2-$\beta$3 loop of RRM3 in PTB34 by Tyr 372 in nPTB34 is conservative and likely to preserve the stacking interaction with the cytosine base at position 5 in CUCUCU RNA oligomer. However, the substitution of Asn 376 (to Ser 377) at the start of the $\beta$3 strand and a triplet of replacements in the $\beta$4-$\beta$5 loop (Asn 413, Gln 421 and Glu 422 in PTB34 to Thr 414, Leu 422 and Asp 423 respectively in nPTB34) may modulate RNA binding (Fig. 4A). Likewise the $\beta$2-$\beta$3 loop of RRM4 in PTB34 is altered in nPTB34 by deletion of Lys 489 and replacement of Arg 491 by His, a pair of changes that may also alter RNA interactions (Fig. 4B). The idea that these structural

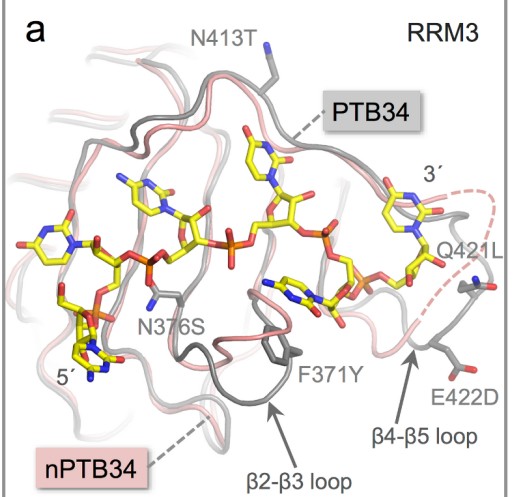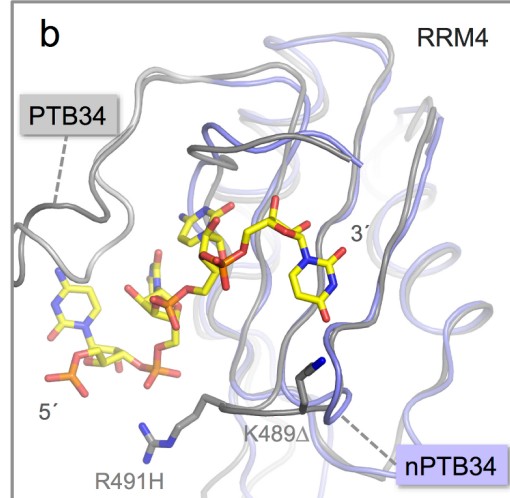

**Figure 4  Location of amino acid differences on the RNA binding surfaces of PTB34 and nPTB34.** (A) Superposition of the solution structure of RRM3 from PTB34 in complex with a hexameric CUCUCU RNA oligomer (PDB ID 2adc; grey) (*Oberstrass et al., 2005*) onto RRM3 from the crystal structure of nPTB34 (this work; pink). The disordered portion of the $\beta4$-$\beta5$ loop in RRM3 of nPTB34 is indicated by a dashed line. The side chains of amino acids that differ between the two structures (discussed in the text) are labeled to indicate the change from PTB34 to nPTB34 and shown for the solution structure of the PTB34-RNA complex. The structural difference in loop $\beta2$-$\beta3$ between the two structures shown most likely reflects conformational variability in this region in the absence of RNA. (B) Superposition of the solution structure of RRM4 with the hexameric CUCUCU RNA oligomer from (PDB ID 2adc; grey) with RRM4 from the nPTB34 crystal structure (this work; light-blue). To avoid cluttering the figure only the four RNA nucleotides that make contact with RRM4 are shown. The side chains of amino acids that differ between the two structures are shown for the substitution R491H and the deletion of Lys 489 (PTB34) in nPTB34, which is labeled $K491\Delta$.

changes affect RNA affinity is supported by mutagenesis experiments which showed that residues in the $\beta2$-$\beta3$ loops of RRMs 3 and 4 and in the $\beta4$-$\beta5$ loop of RRMs are important for RNA binding (*Conte et al., 2000*; *Yuan et al., 2002*). The observed amino acid changes are likely to contribute to the differences observed in the interactions of PTB and nPTB with larger RNA targets (*Pilipenko et al., 2001*). However, it should not be forgotten that sequence differences between PTB and nPTB are focused in regions *outside* the RRM domains and these may well also be important for differences in splicing activity (*Markovtsov et al., 2000*; *Robinson & Smith, 2005*), perhaps by recruiting different protein cofactors.

## MATERIALS AND METHODS

### Plasmid construction and protein expression and purification of nPTB34 for crystal structure determination

The cDNA coding for residues 336–531 of nPTB were amplified by PCR and ligated into the *E. coli* expression vector pETM-11 (*Zou et al., 2003*), which adds an N-terminal hexahistidine tag that could be removed using the Tobacco Etch Virus N1a protease. The fragment was cloned to position the first residue (G336) as the Gly in the TEV protease

recognition site (ENLYFQ/G); protease processing therefore removed all vector-derived amino acids.

Expression of nPTB34 was induced in *E. coli* BL21 (DE3) cells at 37°C for four hours by addition of 1 mM isopropyl $\beta$-D-1-thiogalactopyranoside (IPTG). Cells were resuspended in 0.1 M NaCl, 25 mM Tris HCl pH7.5, 0.1% Triton X-100, 0.5 mM phenylmethylsulfonyl fluoride (PMSF) and 1 mg/ml lysozyme and lysed by sonication. Cell debris and nucleic acid were removed by centrifugation at 25 000 g for 40 min at 4°C in the presence of 1 mg/ml protamine sulfate. nPTB34 was purified from the supernatant by gravity flow through TALON Metal Affinity Resin (Clontech) with stepped elution. The hexahistadine tag was cleaved by addition of 1 mg Tobacco Etch Virus N1a protease for 20 mg of nPTB34 and dialysed overnight into 50 mM Tris pH7.8, containing 0.1 M NaCl, 10% glycerol, 3 mM $\beta$-mercaptoethanol and 0.5 μM ethylenediamine tetraacetic acid (EDTA). Cleaved nPTB34 was purified away from the tag, uncleaved nPTB34 and N1a protease by a second application to TALON resin. Proteins were further purified by size exclusion chromatography on a Superdex 75 (GE Healthcare) using an ÄKTA FPLC system in 25 mM Tris pH 7.8, 0.1 M NaCl and 0.5 mM DTT. Peak fractions were concentrated and stored at −80°C. Expression of Se-Methionine (Se-Met) labeled nPTB34 was performed as above with the following differences: proteins were expressed in *E. coli* B834 (DE3) cells in minimal media supplemented with Se-Met (Molecular Dimensions). Purification proceeded as for unlabeled protein with the addition of 5 mM $\beta$-mercaptoethanol to all buffers. Incorporation of Se-Met was determined to be 100% by MALDI-TOF mass spectrometry (data not shown).

## Crystal structure determination of nPTB34

Purified recombinant nPTB34 was concentrated to ≈5 mg/ml in 100 mM NaCl, 20 mM Tris (pH 7.8) and 0.5 mM DTT. nPTB34 was crystallised by sitting drop vapor diffusion with a reservoir solution containing 0.1 M Tris (pH 8.0), 2 mM $ZnCl_2$ and 20% polyethylene glycol (PEG) 6000. Crystals appeared at first as stacked plates, but single crystals were obtained after micro-seeding into equilibrated drops of nPTB34 in 0.1 M Tris (pH 8.0), 2 mM $ZnCl_2$ and 12% PEG 6000. No crystals were obtained in the absence of $ZnCl_2$—the structure revealed that this is because two crystal–packing interactions are mediated by zinc ions (see Results and Discussion). Crystals were harvested and briefly incubated in mother liquor supplemented with 20% glycerol before flash cooling in liquid $N_2$.

X-ray diffraction data were collected from a single crystal (60 μm × 40 μm × 20 μm) at beamline I02 at the Diamond Light Source (Didcot, UK). Diffraction data were initially indexed in iMosflm and scaled in space group $P2_12_12$ using SCALA (*Collaborative Computer Project No. 4, 1994*) (RRID:nif-0000-30238). Closer inspection of the data in POINTLESS suggested the space group was more likely to be $P2_1$. At this stage molecular replacement was attempted in PHASER (*McCoy et al., 2007*) using the solution structure of nPTB34 as a search model to locate the four molecules in the asymmetric

unit expected from the Matthews co-efficient of 2.08 Da $\text{Å}^{-3}$ but failed to find any acceptable solutions.

We therefore switched to phasing by multi-wavelength anomalous dispersion (MAD) using isomorphous crystals grown from Se-Met labeled nPTB34 (see above). Diffraction data were collected at three wavelengths at beamline I03 at the Diamond Light Source and processed in space-group $P2_1$. Heavy atom site were identified using SHELXD within autoSHARP (*Bricogne et al., 2003*; *Vonrhein et al., 2007*; *Sheldrick, 2008*) and following density modification the figure of merit was 0.802. Initial model building was performed automatically in BUCCANEER (*Cowtan, 2006*) where two nPTB34 molecules were built in the asymmetric unit. We used this initial model in Phaser 2.1 to find the remaining pair of molecules in the asymmetric unit. However, maps calculated using the 'completed' model remained noisy in the region corresponding to this second pair of polypeptide chains and refinement stalled at an $R_{\text{free}}$ value of over 37%. At this stage the data were re-indexed in $P1$ and a molecular replacement search performed using the tetrameric arrangement of crystallographic models generated as described above was used to locate all eight copies of nPTB34 in the larger asymmetric unit. This model was adjusted manually in COOT (*Emsley & Cowtan, 2004*) and refined without any further difficulty. Initially, strict NCS restraints were employed during refinement in CNS v1.2 (*Brunger et al., 1998*). TLS refinement and latter cycles of refinement were performed in Phenix without NCS restraints (*Adams et al., 2010*), yielding an $R_{\text{free}}$ of 27.3%. At this stage the structure was deposited in the Protein Data Bank (PDB; RRID:nif-0000-00135) under ID 4cko, the manuscript published as a preprint (*Joshi et al., 2014*) and submitted for peer review. One reviewer, noting the relatively high value of $R_{\text{free}}$, raised the possibility that the crystals might be twinned, something we had overlooked while resolving the problems due to the initial mis-assignment of the space-group. Analysis using the xtriage function in PHENIX revealed the presence of twinning operators around each of the three principal axes. Re-refinement in REFMAC (*Winn, Murshudov & Papiz, 2003*) (with the presence of 4 twin domains with the following twin operators and fractions: $h, k, l - 0.153$; $-h, -k, l - 0.245$; $h, -k, -l - 0.176$; $-h, k, -l - 0.425$) significantly reduced the values of $R_{\text{work}}$ and $R_{\text{free}}$ (to 17.0% and 20.8% respectively) but did not appreciably improve the electron density map, which was already of high quality. Stereochemistry and clashes were assessed using MolProbity (*Adams et al., 2010*). The final refined coordinates and structure factors have been deposited with the PDB (ID 4cq1).

## Plasmid construction, expression and purification of nPTB34 for solution structure

A 207-residue C-terminal fragment (residues 325–531) of human nPTB (Swissprot Q9UKA9), which contain the third and fourth RRMs, was sub-cloned into a pET-28a(+) vector, using the restriction sites Nde I and Not I. The sequence contained an N-terminal hexahistidine tag. The proteins were over-expressed in BL21(+) *E. coli* cultures grown in M9 minimal medium containing 50 µg/ml Kanamycin. The expression was induced by IPTG at an OD of 0.4. For $^{15}$N-labelling $^{15}$N–NH$_4$Cl was added as the only nitrogen source and for $^{13}$C-labelling normal glucose was replaced by $^{13}$C labeled glucose. The

cell pellets were harvested by centrifugation and resuspended in lysis buffer containing 300 mM NaCl, 50 mM $NaH_2PO_4$ and 1 mM imidazole at pH8 and lysed by 3 passages through a cell-cracker (EmulsiFlex-C5 High Pressure Homogenizer; Avestin, Canada). The protein was purified by two successive Ni-NTA columns (1 ml Ni-NTA resin per liter of culture). Elution occurred between 40 mM and 200 mM imidazole. After the second column the protein was dialyzed against the NMR buffer containing 20 mM NaCl and 10 mM $NaH_2PO_4$ at pH 5.8 and concentrated by centricon ultra centrifugation to 1 to 2 mM. The purity of the protein sample was checked by SDS PAGE. The final yield per liter of culture was estimated to be around 20 mg protein. Mass spectroscopy was used to characterize the proteins.

## NMR spectroscopy

Shigemi NMR tubes with 250 µl sample solution (10% or 100% $D_2O$) were used. NMR spectra were acquired at 303 K on a Bruker DRX-600 and a Avance-900 spectrometer. NMR data were processed using Topspin 3.0 (Bruker) and analyzed using Sparky (*Goddard & Kneller, 2007*).

For the sequence specific backbone assignment HNCA, HN(CO)CA, CBCA(CO)NH, HNCACB, $^{15}N$ HSQC, $^{13}C$ HSQC as well as $^{15}N$ TROSY spectra were recorded. The aliphatic side chains were assigned based on a $^{15}N$ NOESY and a $^{13}C$ NOESY with the help of a $^{15}N$ TOCSY and a HCCH-TOCSY. The assignment of side chain amides was achieved by analyzing the $^{15}N$ NOESY and the $^{15}N$ HSQC, while for the assignment of the aromatic side chains a 2D TOCSY and a 2D NOESY in $D_2O$ were recorded. Distance restraints used in structure calculations were extracted from $^{15}N$-NOESY-HSQC and $^{13}C$-NOESY-HSQC in $H_2O$ and 2D NOESY in $D_2O$. Hydrogen-bonded NH groups were identified by the presence of amide resonances in $^{15}N$ HSQC spectra, which were recorded immediately after lyophilizing and dissolving the sample in $D_2O$.

## NMR structure calculation

Initial peak picking and nOe assignments were performed using the ATNOSCANDID package (*Herrmann, Guntert & Wuthrich, 2002a*; *Herrmann, Guntert & Wuthrich, 2002b*). Peak lists of the final seventh cycle were used as an input for the program CYANA 3.0 (*Guntert, 2004*). The "noeassign" protocol of CYANA was used to reassign and calibrate the nOe signals of the given peak lists. These lists were manually reviewed and the distance restraint list were further used by CYANA to calculated 250 structures by a simulated annealing protocol (20 000 MD steps). Based on the target function the 50 best structures were selected for refinement using a simulated annealing protocol with the AMBER 9 suite (*Pearlman et al., 1995*) and against the ff99 force field (*Lindorff-Larsen et al., 2010*) using implicit water (*Bashford & Case, 2000*). The 20 final conformers were selected using a combined AMBER energy and violation energy as selection criteria. The coordinates have been deposited in the Protein Data Bank (ID 2mju); NMR chemical shifts and distance constraints used in the structure calculation have been deposited under ID 19737 to the Biological Magnetic Resonance Data Bank (BMRB; RRID:nif-0000-21058).

## ACKNOWLEDGEMENTS

We thank staff at the Diamond Light Source (Didcot, UK) for assistance with X-ray data collection. We are grateful to Reviewer 3 for drawing our attention to the possibility that the diffraction data might be twinned.

### Funding

This work was supported in part by grant support from the Biotechnological and Biological Sciences Research Council (BBSRC) to SC and from SNF-NCCR Structural Biology to FHTA. AJ is grateful for the award of a BBSRC PhD studentship. The funders had no role in study design, data collection and analysis, decision to publish, or preparation of the manuscript.

### Grant Disclosures

The following grant information was disclosed by the authors:
Biotechnological and Biological Sciences Research Council (BBSRC).
SNF-NCCR Structural Biology.

### Competing Interests

We declare that we have no competing interests relevant to this publication.

### Author Contributions

- Amar Joshi and Markus Blatter conceived and designed the experiments, performed the experiments, analyzed the data, contributed reagents/materials/analysis tools, wrote the paper, prepared figures and/or tables, reviewed drafts of the paper.
- Vicent Esteve conceived and designed the experiments, performed the experiments, analyzed the data, contributed reagents/materials/analysis tools, wrote the paper, reviewed drafts of the paper.
- Adrian N. Buckroyd performed the experiments, contributed reagents/materials/analysis tools, wrote the paper, reviewed drafts of the paper.
- Frédéric H.-T. Allain and Stephen Curry conceived and designed the experiments, analyzed the data, wrote the paper, prepared figures and/or tables, reviewed drafts of the paper.

### Data deposition

The following information was supplied regarding the deposition of related data:

Coordinates, X-ray structure factors and NMR assignments used in structure determination have been deposited with the Protein Data Bank. nPTB34 crystal structure and structure factors: PDB ID - 4cq1. nPTB34 NMR solution structure - PDB ID 2mju. nPTB34 NMR chemical shifts and distance restraints have been deposited in the BMRB under ID 19737.

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
