# Peer review of "Solution and crystal structures of a C-terminal fragment of the neuronal isoform of the polypyrimidine tract binding protein (nPTB)"

_PeerJ, doi:10.7717/peerj.305_

## Round 0.1 · original submission · Minor Revisions

As you can see, your work generated very positive response. Some minor corrections were suggested by two reviewers and these critical points need to be addressed in the revised manuscript.

·

Basic reporting

1) Fig 4a shows that the conformation of one loop (labelling of the loop on the figure would help, if possible) is different between PTB34 and nPTB34. Do the Authors believe that such difference is significant or is its effect mitigated by loop flexibility? Are there residues that are conserved in the two proteins but, due to the different conformation, are proposed to be in contact with the RNA only in one of them?

2) Line 113: “55 hydrogen bond restraints identified from slow exchanging amides”
How where the slow exchanging amides identified? I would add a couple of words to describe this, if possible.

3) Line 52: βαββαβtopology ; space seems to be missing

4) Line 229: “Peak fractions were concentrated and stored at 80°C.”
Was it -80, perhaps?

5) Line 288: “Shigemi NMR tubes with 250 L”
The symbol for microliters is missing in the pdf version I downloaded

6) Figure 4 caption
“side chain of the F317Y substitution is disordered in the crystal structure”

Residue F371Y is shown in the picture. Is it a typo in one of the two cases?
Also, I would like to see the sidechains (sticks) of F317Y and N376S in figure 4, if possible. The former would make it easier to visualize the proposed stacking with the RNA bases, for instance.

Experimental design

no comments

Validity of the findings

no comments

Comments for the author

The Authors report the x-ray and NMR structure of a fragment of nPTB protein. The article is clear and concise and the experiments are well conducted. Although the reported structure is very similar to those already available for the homologous PTB protein, as expected, reading the manuscript is still interesting.
I believe that the manuscript can be published with very minor alterations as indicated in the "basic reporting" section.

Reviewer 2 ·

Basic reporting

The submission adheres to all PeerJ policies.
The structure of the submitted article does not conform to common template: Introduction, Results and Discussion, Materials and Methods
Some references are not complete: see, for example, Llorian et al.
Some references in Reference list have no full title of the journal.

Experimental design

No comments

Validity of the findings

No comments

Comments for the author

No comments

Reviewer 3 ·

Basic reporting

No Comments

Experimental design

No Comments

Validity of the findings

No Comments

Comments for the author

The manuscript with title of “Solution and crystal structures of a C-terminal fragment of the neuronal isoform of the polypyrimidine tract binding protein (nPTB)” determined the structure of neuronal PTB C-terminal RRM 3 and RRM4 with X-ray crystallography and NMR method. The structure of nPTB34 provides structural evidence that shows RRMs 3 and 4 have fixed arrangement, a conserved feature in PTB homologs. By superimpose nPTB34 to PTB34 or RRM2-Raver1 complex structure, the authors show the interface between RRMs or RRM and co-regulator protein is conserved. The authors further pinpoint the residues which are responsible for specificity of RNA binding. The manuscript is well organized and I suggest accept with minor revision.

Below are questions and suggestions:
1. As for crystal structure determination of nPTB34, the authors determined the structure by lower the symmetry from P21212 to P1, but the final Rfree is still quite high (27.4%) considering the high resolution (1.7 angstrom). Have the authors check other problems with diffraction data, such as twin, etc?
2. In the part of Crystal structure of nPTB34, the authors use one paragraph to describe the zinc ion which is critical for crystal packing. Without a figure, I feel difficult to follow. In fact, I don’t think is necessary to spend many words on zinc, since it’s only a crystal artifact and have no physiological function.

---

## Round 0.2 · accepted · Accept

I would like to thank you for addressing all the critical points of reviewers and for the careful revison of the manuscript.